# Neonicotinoid Microsphere Immunosensing for Profiling Applications in Honeybees and Bee-Related Matrices

**DOI:** 10.3390/bios12100792

**Published:** 2022-09-26

**Authors:** Mang Xu, Liza Portier, Toine Bovee, Ying Zhao, Yirong Guo, Jeroen Peters

**Affiliations:** 1Wageningen Food Safety Research, Akkermaalsbos 2, 6708 WB Wageningen, The Netherlands; 2Institute of Pesticide and Environmental Toxicology, Key Laboratory of Biology of Crop Pathogens and Insects of Zhejiang Province, Zhejiang University, Hangzhou 310058, China; 3Ministry of Agriculture Key Laboratory of Molecular Biology of Crop Pathogens and Insects, Zhejiang University, Hangzhou 310058, China

**Keywords:** neonicotinoids, microspheres, immunoassay, bees, pollen, honey, surface water

## Abstract

Neonicotinoids are the most commonly used insecticides due to their effectiveness. However, non-targeted insects, especially bees, are also affected by neonicotinoids. Therefore, neonicotinoid application can contribute to the declining bee populations worldwide. The presented study describes the development of novel competitive, fluorescent microsphere-based suspension immunoassays for neonicotinoid profiling and their application to bees and essential bee-related matrices, using the Multi-Analyte Profiling (xMAP) technology. For the construction of these neonicotinoid microsphere immunoassays (nMIAs), neonicotinoid–ovalbumin conjugates were coupled to unique fluorescent, paramagnetic microspheres, which competed with the free neonicotinoids that were present in test samples for interacting with the corresponding, specific antibodies. In total, five independent nMIA’s were developed for the detection of imidacloprid, acetamiprid, clothianidin, thiacloprid, thiamethoxam, dinotefuran, nitenpyram and imidaclothiz with the limits of detection being for 0.01 ng/mL, 0.01 ng/mL, 0.02 ng/mL, 0.02 ng/mL, 0.003 ng/mL, 2.95 ng/mL, 0.09 ng/mL and 0.04 ng/mL, respectively. The developed nMIAs were applied to fortified matrices including surface water, pollen, honey and honeybees. All of the neonicotinoids, except dinotefuran, could be sensitively detected in all of the tested environmental matrices and bees, with there being sensitivities of 1 ng/mL in water and 10 ng/g in solid materials. These nMIAs provide a rapid profiling method for all of the common neonicotinoids, including those that are banned by the European Union for outdoor use. The developed method can contribute to healthy and sustainable beekeeping, globally, via its application in the apiary environment.

## 1. Introduction

Honeybees (*Apis mellifera*) are considered to be major pollinators [1]. Research over the last decade indicates that the populations of honeybees are decreasing, in some areas even up to 96% [2], and several studies have suggested that the heavy usage of neonicotinoid pesticides are responsible for the honeybee population decline [3,4,5]. Neonicotinoids are a group of effective synthetic insecticides (Table 1, Figure 1) that were synthesized for the first time in the 1980s and introduced to the market in 1991 in the form of imidacloprid [6]. Due to their high toxicity to insects and the little effect that they have on mammals [7], neonicotinoids soon became the most used insecticide class in agriculture, taking nearly a third of the global market share in 2011 [8]. Unfortunately, neonicotinoids also have serious detrimental effects on non-targeted organisms like bees [9].

It is suggested that both the direct and systemic exposure of honeybees to neonicotinoids has led to increased mortality amongst bees [7,9]. Although several studies have showed that neonicotinoids including imidacloprid, clothianidin and thiamethoxam are not lethal to bees at field-realistic concentrations, the collective exposure to these neonicotinoids can still affect the bees’ health, stress levels and homing ability [14,15,16]. Bees are mainly exposed to neonicotinoids by them coming into contact with contaminated plants while foraging, and from there, neonicotinoids can be further transferred to honey, beebread and beeswax [17,18,19,20].

Neonicotinoids are very stable and persist in soil and water for months to years after their application on crops, which means that they accumulate and that their environmental impacts continue for a considerable period of time [21]. Due to their serious environmental impact, risk assessments of neonicotinoids were conducted by the European Food Safety Authority (EFSA), and the use of imidacloprid, thiamethoxam, and clothianidin was restricted by the European Union (EU). A temporary ban was imposed for their application on flowering crops outside of greenhouses in 2013 and this ban was prolonged in 2018 [22,23,24]. In 2020, the approval of thiacloprid was discontinued based on the conclusion that was drawn by EFSA [25]. Maximum residue levels (MRLs) were set for the five neonicotinoids that are approved for use within the EU, namely imidacloprid, acetamiprid, clothianidin, thiacloprid and thiamethoxam (ranging from 0.01 to 50 mg/kg) [26]. Additionally, some plant trading companies, as a part of their corporate social responsibility (CSR), have clearly defined several neonicotinoids as unwanted pesticides on their plant products (e.g., pyrethroids, fipronil and chlorpyrifos). In addition, retailers who acquire plants from those plant trading companies have their own CSR demands considering pesticides which are harmful to bees and other insects. Meanwhile, increasing the consumers’ awareness has made the retailers more selective, leading to product lines that are labelled “certified free from neonicotinoids” (personal communication).

Various detection methods have been developed and can be implemented to control the enforcement of these EU restrictions, screen agricultural products or control the foraging environment of the honeybees for the presence of neonicotinoids. One of the commonly used detection methods for neonicotinoids is the instrumental analysis: liquid chromatography which is paired with tandem mass spectrometry (LC-MS/MS). LC-MS/MS multi-methods were developed for detecting up to eight neonicotinoids and some of their metabolites in bee-related matrices including pollen, honey and bees with quantification ranges of 0.1–0.5 µg/kg [18,27]. Despite their high sensitivity and multi-analyte approach, these instrumental analyses are usually time-consuming, expensive, require extensive sample clean-up and need to be operated by highly skilled personnel. The electrochemical and fluorescent-based detection of neonicotinoids has also been exploited. Zhai et al. reported an electrochemical monitoring platform using β-cyclodextrin-reduced graphene oxide/glassy carbon electrodes for thiamethoxam detection in brown rice, thereby reaching a sensitivity of 78.8 ng/g, yet this device requires strict temperature stabilization [28]. Srinivasan et al. developed a Cu-rGO composite nanofibers modified glassy carbon electrode electrochemical sensor method for the detection of imidacloprid in soil samples, with the limit of detection being as low as 0.642 ng/g [29]. El-Akaad et al. reported a reusable molecular imprinted polymer-based sensor for imidacloprid detection which showed selectivity to five neonicotinoids in water samples, however, the sensitivities were only 1.2 µg/mL in the water samples [30]. A Quenchbody-based assay was developed for the detection of imidacloprid at 10 ng/mL [31], while Yang et al. developed a fluorescent aptamer-based aggregation assay which could detect 24 ng/g acetamiprid in celery leaves and Chinese green tea [32]. Recently, Wang et al. reported an imprinted fiber array strategy which was able to detect imidacloprid, dinotefuran, acetamiprid, nitenpyram and thiamethoxam at low ng/mL levels [33].

As a faster and easier approach, immunoassay-based methods can also be considered as biosensors for the detection of neonicotinoids. Enzyme-linked immunosorbent assays (ELISA) have been developed to detect imidacloprid in food samples at ng/mL levels, with there being a slight cross-reactivity for a few other neonicotinoids [34]. Commercial ELISA test kits for screening for various neonicotinoids are also available on the market [35]. Paper-based immunochromatographic lateral flow devices (LFDs) have been developed to screen for traces of neonicotinoids. Sensitive LFDs for imidacloprid and acetamiprid showed a high sensitivity of 0.5–1 ng/mL and a broad detection [36,37]. Besides the aforementioned LC-MS/MS multi methods, other platforms are hardly used to screen for all of the eight common neonicotinoids. The microsphere-based Multi-Analyte Profiling (xMAP) technology offers simple, multiplex and semi-high throughput screening possibilities and it has been previously used for the detection of pesticides and mycotoxins, etc. [38,39]. In an xMAP assay, various types of biological reagents can be immobilized on the surface of the paramagnetic microspheres (often referred to as beads) in a solution, while only the well surface of the microtiter plate serves as an immobilizing area for antigens in ELISAs. Therefore, the diffusion distance between the analyte and the sensor elements, namely the antibodies, is shorter in xMAP compared to in ELISA [38,39].

In this study, five single neonicotinoid microsphere immunoassays (nMIAs) were developed using xMAP technology, which enabled the detection of eight neonicotinoids, namely imidacloprid, acetamiprid, clothianidin, thiamethoxam, thiacloprid, dinotefuran, nitenpyram and imidaclothiz. Most importantly, these assays sensitively detected all five of the neonicotinoids that are currently banned by the EU and another three neonicotinoids which are not released to the European market. The developed assays were applied to matrices that were relevant to the honeybee environment and to honeybees directly. Simple and straightforward hot water extractions were implemented, and this resulted in the successful detection of all eight neonicotinoids in all of the matrices at low ng/mL levels. The combination of these assays allows for the total profiling of all eight of the common neonicotinoids in bee-related matrices. The conducted research is part of the European H2020 B-GOOD project (https://b-good-project.eu/) that develops new diagnostic tools that help to map the health status of honeybee colonies and their environment to give beekeeping guidance by computational-assisted decision making. Therefore, we chose to focus on essential resources for bees (water and pollen) and matrices that are relevant to function as health-based indicators of the bee environment (bees and honey).

## 2. Materials and Methods

### 2.1. Instruments

A MAGPIX planar array analyzer that was equipped with xPONENT 4.3 software (Luminex, Austin, TX, USA) was used for the readout of the nMIAs. A DynaMag-2 magnetic particle concentrator (Invitrogen, Oslo, Norway) and a Life Sep 96F magnetic plate (Sigma Aldrich, Zwijndrecht, the Netherlands) were used for microsphere coupling procedure and concentrating the paramagnetic microspheres during the assay preparation, respectively. An Eppendorf 5810R centrifuge that was equipped with an A-4-62 rotor (VWR, Amsterdam, the Netherlands) was used for centrifuging the filter plates (supplier). An Edmund Bühler GmbH plate shaker (Salm en Kipp, Breukelen, The Netherlands) was used during the incubation steps of the assay. A homogenizer for grinding the bees was purchased from Ystral Gmbh (Dottingen, Germany). Folded Whatman filter paper (595 ½) was used for filtering the extracts and it was purchased from GE Healthcare (Solingen, Germany). GraphPad Prism version 8.0.0 for Windows was used for the curve fitting of the data.

### 2.2. Chemicals and Reagents

For the development of the nMIAs, the MagPlex MC10066, MC100064, MC10038 and MC10052 paramagnetic microspheres (Luminex; Austin, TX, USA) were used for coupling the neonicotinoid–ovalbumin (OVA) conjugates. The MAGPIX system runs on drive fluid (Luminex; Austin, TX, USA). All of the immunoassay steps were performed in CELLSTAR^®^ 96-wells culture microtiter plates (Greiner, Alphen a/d Rijn, The Netherlands). Sulfo-NHS (N-hydroxysulfosuccinimide), EDC (1-ethyl-3-(3-dimethylaminopropyl) carbodiimide and caffeine-(trimethyl-13C3) were purchased from Sigma-Aldrich (Zwijndrecht, the Netherlands). All of the mouse monoclonal antibodies (mAbs) against each neonicotinoid target and the neonicotinoid–OVA conjugates (Appendix A) were kindly supplied by the Institute of Pesticide and Environmental Toxicology of Zhejiang University (IPET-ZJU). Goat anti-mouse IgG that was coupled to R-phycoerythrin (GaM-RPE) was used as the reporter molecule (Moss Inc, Maryland, USA). All of the solid chemicals for buffer preparation were purchased from Sigma Aldrich (Zwijndrecht, The Netherlands), except for monosodium phosphate, which was purchased from Merck (Darmstadt, Germany). Imidacloprid (purity: 98.7%), thiamethoxam (purity: 99.9%) and thiacloprid (purity: 99.9%) solid standards were purchased from HPC Standards GmbH (Borsdorf, Germany), while acetamiprid (purity: 99.9%), clothianidin (purity: 99.9%) and nitenpyram (purity: 98.6%) were ordered from Sigma-Aldrich (Zwijndrecht, The Netherlands). Dinotefuran (purity: 97.5%) and imidaclothiz (99%) standards were purchased from Dr Ehrenstorfer GmbH (Augsburg, Germany).

Pollen was purchased from het Bijenhuis (Wageningen, the Netherlands). Blank honeybees were kindly supplied by Coby van Dooremalen (Wageningen Plant Research, the Netherlands). The blank honey was kindly supplied by Nuno Capela of the University of Coimbra, Portugal. The blank water samples for the method validation were previously collected by Dr Rubing Zou from the Institute of Pesticide and Environmental Toxicology of Zhejiang University (Appendix A) [40]. All of the aforementioned blank samples were confirmed to be free from neonicotinoids by an in-house LC-MS/MS neonicotinoid multi-method analysis (results not shown).

### 2.3. Coupling of Neonicotinoid–OVA Conjugates to Paramagnetic Microspheres

The neonicotinoid–OVA conjugates were independently coupled to the surface of a unique microsphere set according to the existing protocols that relate to applying 1-Ethyl-3-(3-dimethylamino propyl) carbodiimide/N-hydroxysuccinimide coupling chemistry [41]. In short, the following conjugate-microsphere pairs were chosen: imidacloprid-OVA:MC10066, acetamiprid-OVA:MC10064, clothianidin-OVA:MC10012, thiacloprid-OVA:MC10038 and thiamethoxam-OVA:MC10052 for the imidacloprid assay (Imi assay), acetamiprid assay (Ace assay), clothianidin assay (Clo assay), thiacloprid assay (Thc assay) and thiamethoxam assay (Thm assay), respectively (Appendix A). In total, 2.5 × 10^6^ microspheres of each microsphere set were activated by incubation with EDC/Sulfo-NHS. Next, the OVA–neonicotinoid conjugates were coupled to microspheres by suspending the unique microspheres in 0.5 mL of a 0.1 mg/mL target conjugate solution. The coupled microspheres were stored in 1x PBST (PBS containing 1% BSA, 0.02% Tween-20 and 0.05% NaN₃, pH 7.4) at 4 °C in the dark until their use. Before each use, the microspheres were briefly but thoroughly vortexed into homogeneous suspensions.

### 2.4. Antibody Dilution Factor Optimization

To determine the optimal concentration of each antibody for the respective assay, a series of antibody concentrations ranging from 1 ng/mL to 10 µg/mL were prepared via 2-fold stepwise serial dilution in PBST-BSA (PBS with 0.1% BSA and 0.05% Tween-20, pH 7.4). These antibody solutions were incubated with 1000 neonicotinoid–OVA conjugates that were coupled with microspheres for 20 min in the dark, while shaking them at 400 rpm. The excess antibodies were removed by washing the solutions with PBST-BSA. One hundred µL of two µg/mL GaM-RPE in PBST-BSA was added to the microsphere–antibody complex and incubated as described previously. The excess GaM-RPE was removed by washing it with PBST-BSA. Finally, the microspheres were resuspended in PBST-BSA on the plate shaker, and the reporter signal was measured using the MAGPIX planar array analyzer, with the microsphere count was set at 50 for each assay. Output data were reported as median fluorescent intensity (MFI). The optimal working concentration of each antibody was determined corresponding to an MFI reading of approximately 1000.

### 2.5. Sensitivities of the nMIAs

To determine the sensitivities of the nMIAs, a neonicotinoid calibration standard series for imidacloprid, acetamiprid, clothianidin, thiamethoxam and thiacloprid, ranging from 1 µg/mL to 0.1 pg/mL, were prepared in PBST-BSA via 10-fold stepwise serial dilution from the stock standards (100 µg/mL in Milli-Q water). A PBST-BSA buffer without fortifications was used as a negative control in each assay. Next, the neonicotinoid dilution series (100 µL) were incubated with the corresponding microspheres (10 µL for approximately 1000 microspheres) and antibodies at their optimal assay concentrations (10 µL), for 20 min in the dark while they were shaken. The assay was performed in triplicate and analyzed as described in Section 2.4. Thus, five single dose-response curves for the target neonicotinoids were generated for each microsphere immunoassay (Appendix A).

### 2.6. Determination of Cross-Reactivity

To investigate the cross-reactivity of all of the available neonicotinoids within the five nMIAs, additional calibration standard series were prepared for imidaclothiz, nitenpyram and dinotefuran by 10-fold stepwise serial dilutions in PBST-BSA, each with a calibration standard range being 1 µg/mL to 0.1 pg/mL. All of the prepared calibration standards were measured in the five nMIAs in duplicate. The protocol was the same as is mentioned in the Section 2.5. Four-parameter (asymmetric) logistical models were applied to construct the dose-response curves. The cross-reactivity was determined by comparing the inhibiting concentrations at 50% of the maximum response (IC_50_) of the target neonicotinoids with the IC_50_ values of all of the cross-reacting neonicotinoids.

### 2.7. Preliminary Matrix Validation of the nMIAs

One g of material (tap water, pollen, honey or bees) was fortified with individual neonicotinoids at 100 ng/g and 10 ng/g using stock solutions. After fortification, the solid samples were air-dried for 30 min at room temperature. The fortified solid samples were extracted by adding 10 mL of hot tap water, followed by a brief period of them being manually shaken. Bee samples were homogenized with a homogenizer immediately after the addition of hot water. Water samples were directly diluted at 1:10 with tap water. After cooling the extracts of the solid material to room temperature, the extracts were filtered through filter paper and then through a 0.2 µm filter plate. For the analysis, the microsphere suspensions were prediluted at 1:100 in 10× concentrated PBST-BSA (10× concentrated PBS containing 1% BSA and 0.5% Tween-20, pH 7.4), while the antibodies were prediluted to 10× their optimal concentrations in regular PBST-BSA. Ten µL of microsphere solution and 10 µL of antibody solution were added to 90 µL of the extracts to constitute the assay in 1× PBST-BSA. The assays were performed in duplicates as described in previously.

### 2.8. Preliminary Validation of the Robustness of the nMIAs

To test the robustness of the developed nMIAs, we followed a validation protocol for screening assays [42]. To this end, individual blank surface water samples, which were collected from waterbodies (Appendix A), were fortified with 5 ng/mL imidacloprid, thiacloprid, clothianidin or thiamethoxam for their analysis in the Imi assay, Thc assay, Clo assay and Thm assay, respectively. Sufficient stock solutions of microspheres and antibodies were prepared in triplicate over 3 days for the measurement of the 20 blank and 20 fortified samples. All of the prepared solutions were stored in the dark at 4 °C until their use. Ninety µL of each blank and fortified water sample were independently incubated with the antibody and microsphere solution and developed as described in Section 2.7. Both the blank and fortified water samples were measured in parallel over three consecutive days.

## 3. Results and Discussion

### 3.1. Assay Optimization

Neonicotinoids are small molecules (haptens) that require them to be coupled to larger carrier proteins in order to be used as coating antigens in immunoassays. Both OVA and BSA are commonly used for the generation of these hapten–protein conjugates. Based on antibody–antigen interactions, imidacloprid, acetamiprid, clothianidin, thiacloprid and thiamethoxam were selected as the most suitable hapten conjugates for their use in the profiling assays (Appendix A). The principle of the nMIAs is explained in Figure 2.

The selected conjugates were successfully coupled to the carboxylated surfaces of the microspheres, as was determined by a maximum MFI response test. After the max response test, the optimal dilution factors for the primary antibodies were determined for each assay, aiming at a maximum MFI of 1000 (results not shown).

### 3.2. Assay Sensitivities

The initial aim of the presented research was to develop a multiplex method for the detection of all the neonicotinoids in one single measurement. However, due to the high similarity of the chemical structures of the neonicotinoids (Figure 1), significant cross-interactions were observed when all of the conjugate-coupled microspheres were incubated as an array with each single target antibody in a single well (Appendix A). Therefore, five single assays for each neonicotinoid target were used for further evaluation. To demonstrate the detection sensitivity for each assay, five separate neonicotinoid calibration standards were prepared for imidacloprid, acetamiprid, clothianidin, thiacloprid and thiamethoxam to create the dose-response curves. The five dose-response curves for each assay are shown in Figure 3.

The B/B_0_ values indicate the inhibition levels of the target neonicotinoid at each standard concentration, which is calculated by normalizing its MFI with the maximum MFI signal that is obtained from measuring the negative control (PBST-BSA). In all of the assays, the B/B_0_ increased as the neonicotinoid concentration decreased in the dynamic range. The limit of detection (LOD) was set at the average MFI value of the negative controls after subtracting three times its standard deviation, while the IC_50_ of each assay was determined at 50% of the maximum B/B_0_. The dynamic measurement range for each target was arbitrarily set from 20% to 80% inhibition (IC_20_-IC_80_). These parameters were calculated via interpolation from the four-parameter logistics of the dose-response curves. All five nMIAs showed adequate sensitivities, which resulted in LODs of 0.01 ng/mL, 0.02 ng/mL, 0.19 ng/mL, 0.02 ng/mL and 0.003 ng/mL for the Imi assay, Ace assay, Clo assay, Thc assay and Thm assay, respectively (Appendix A). A comparison of the obtained LODs for all eight of the neonicotinoids in the five nMIAs to those that were obtained in the other neonicotinoid detection (immuno)assays showed that the nMIAs are generally more sensitive than the LFIAs, ELISAs and other immunoassay formats are, and that they are substantially more sensitive than the electrochemical-, molecular imprinted polymer and aptamer-based assays are (Table 2). A comparison to the closely related ELISA-based formats shows that there are 10- to 100-fold higher sensitivities for the nMIAs. The workload of the nMIAs and ELISA-based assays is largely comparable, as are the incubation times. The LFIAs, on the other hand, are more rapid but not suitable for a semi-high throughput analysis. For the nMIAs, the preparation and measurement of the 96 samples was achieved within 2 h.

### 3.3. Cross-Reactivity Testing

To evaluate the possibility to screen for the presence of all of the common neonicotinoids that are used in global agricultural production, all eight of the neonicotinoid calibration standard series were measured individually in the five nMIAs. The respective cross-reactive dose-response curves for these eight neonicotinoids in the five nMIAs are depicted in Appendix A, while all the cross-reactivity data are shown in Table 3.

The results show that the Imi assay can detect all of the neonicotinoids, but it can only detect thiamethoxam and dinotefuran at high concentrations. The Thc assay is most sensitive for thiacloprid and acetamiprid. Besides acetamiprid, the Ace assay can also detect thiacloprid, while the Clo assay can only detect clothianidin and dinotefuran. The Thm assay highly specifically detects thiamethoxam. These results show that all eight of the most common neonicotinoids can be sensitively detected by at least one of the nMIAs with IC_50_ values of 0.07 ng/mL, 0.019 ng/mL, 1.9 ng/mL, 0.06 ng/mL, 0.01 ng/mL, 5.8 ng/mL, 0.67 ng/mL and 0.11 ng/mL for imidacloprid, acetamiprid, clothianidin, thiacloprid, thiamethoxam, dinotefuran, nitenpyram and imidaclothiz, respectively. This indicates that the developed assays show good potential for the sensitive, broad-spectrum profiling of neonicotinoids in practical samples.

### 3.4. Application of the NMIA to Surface Water

Neonicotinoids are polar molecules that are soluble in water and therefore, they are easily spread through the environment. With surface water being an important factor in the bee environment, we fortified tap water with the eight individual neonicotinoids at concentrations of 1 ng/mL and 10 ng/mL. The fortified tap water samples were further diluted 10-fold with tap water to anticipate the presence of future surface water samples that may contain disturbing matrix contents. The fortified water samples, that were diluted in blank tap water, were measured in all five of the nMIAs. Blank tap water was also used as a blank control. The results, which are displayed in the inhibition-based graphs, are shown in Figure 4. For each assay, the cut-off level for a screen-positive detection was set at 20% inhibition. All the fortified neonicotinoid samples were detected in one or more nMIAs. The best-achieved sensitivities for all of the tested neonicotinoids in water were 1 ng/mL for imidacloprid with the Imi assay, acetamiprid with the Ace assay, thiacloprid with the Thc assay, thiamethoxam with the Thm assay and dinotefuran with the Clo assay; 10 ng/mL for clothianidin with the Imi assay, nitenpyram with the Imi assay, and imidaclothiz with the Imi assay (Appendix A). Nevertheless, neonicotinoids that are found in large, natural waterbodies are usually present at trace levels due to there being large volumes and movements. For instance, Mahai et al. applied LC-MS/MS to screen water from the central Yangtze River, China, and they reported that there were various levels of imidacloprid, acetamiprid, thiamethoxam, nitenpyram, clothianidin and thiacloprid contaminations with the medians of 4.37 ng/L, 2.50 ng/L, 1.10 ng/L, 0.34 ng/L, 0.10 ng/L and 0.02 ng/L, respectively [50]. Trace-level concentrations of neonicotinoids like these are unfortunately undetectable with nMIAs unless preconcentration steps are implemented.

### 3.5. Pre-Validation of the nMIAs for Surface Water Samples

Previously collected global surface water samples from ten different locations [40] were confirmed to be free from neonicotinoid contaminations by an LC-MS/MS analysis (data not shown). To prove that the developed nMIAs are fit for purpose, we chose to validate them at ½ of the lowest MRL that is legislated in the EU Pesticides Database with 5 ng/mL in water [26] using guidelines for the validation of screening methods for the residues of veterinary medicines [42]. Additionally, this low level of fortification for the validation is desirable as all of these neonicotinoids have been banned for outdoor use besides acetamiprid. Due to the overlap in the detection characteristics between the Thc assay and Ace assay, it was decided that the latter was not crucial for screening, and therefore, it was not included in the presented validation. Next, the ten true blank water samples were fortified in duplicate and these 20 fortified and 20 blank samples were analyzed in the nMIAs, which was divided over three consecutive days. The relative responses for both the blank and fortified water samples were calculated by dividing all of the absolute responses of each sample by the response of one single blank sample which was measured on each day. According to the guidelines for the validation of screening methods [42], we calculated the threshold (T) and cut-off factor (Fm). The threshold was calculated by the following formula:Threshold (T) = B − 1.64 × SDb

In this formula, B is the mean response of all the blank control samples, while SDb is the standard deviation of these blank responses. The cut-off factor was calculated by the next formula:Cut-off factor (Fm) = M + 1.64 × SDm

In this formula, M is the mean response for all of the fortified samples, while SDm is the standard deviation of these responses. The independent B/B_0_ values that were obtained for the blank control and fortified control samples were plotted in graphs, together with the threshold value and cut-off factor for each assay (Appendix A). The outcomes demonstrate that the highest relative response of all of the blank samples is significantly lower than all of the relative responses of the fortified sample, indicating the absence of false compliant results. All of the nMIA screening assays showed Fm < B and Fm < T, meaning that all of the nMIAs were successful as screening assays for their target neonicotinoids at 5 ng/mL levels. Even though the water samples originated from very diverse water bodies, the validation results for the nMIAs (Appendix A) show that the compositional differences (e.g., minerals) did not influence the assays at the critical concentrations that were assessed.

### 3.6. Application of the nMIAs to Other Bee-Related Matrices

Neonicotinoids are highly soluble in water and can be easily extracted from matrices using water [31]. In order to simplify the sample preparation of these screening assays to a more on-site-friendly scenario, hot tap water was used for the sample extraction of the pollen, honey and bee matrices in this study. All of the matrices were fortified with 100 ng/g and 10 ng/g of every neonicotinoid, which is equivalent to 10 ng/mL and 1 ng/mL, respectively, after the tap water extraction was performed. Appendix A shows that the pollen, honey and bee samples resulted in identical profiling patterns for each single neonicotinoid in the five nMIAs. Eventually, all of the neonicotinoids could be detected in these three matrices: imidacloprid in the Imi assay at 10 ng/g, acetamiprid in the Ace and Thc assays at 10 ng/g, clothianidin in the Imi assay at 10 ng/g, thiacloprid in the Ace and Thc assays at 10 ng/g, thiamethoxam in the Thm assay at 10 ng/g, dinotefuran in the Clo Assay at 100 ng/g, nitenpyram in the Imi assay at 10 ng/g and imidaclothiz in the Imi assay at 10 ng/g (Table 4). Nitenpyram could only be detected by the Imi assay, as it was expected, while for dinotefuran, we observed unexpected inhibitions in the pollen and bees at 10 ng/mL. Therefore, the detection of nitenpyram in both the pollen and bees was only reliable in the Imi assay, while nitenpyram could not be detected in the honey. The discrepancies in detection ability when the nMIAs were applied in different matrices might arise from the matrix effect. Thus, a careful matrix validation is important before the actual application of the nMIAs in any matrix. Nonetheless, the developed assays meet the validation criteria and prove to be a promising method for the broad-spectrum screening of neonicotinoids in different matrices. In 2016, David et al. reported the occurrences of thiacloprid (78 ng/g), thiamethoxam (11 ng/g) and clothianidin (11 ng/g) in rapeseed flower pollen as determined by an LC-MS/MS analysis [51]. If they are applied to these pollen samples, the nMIAs are likely able to detect the presence of thiacloprid and thiamethoxam but would not be sensitive enough to detect clothianidin based on the inhibition profiles in Appendix A. In another study, Codling et al., focused on neonicotinoid contaminations in honey, pollen and honeybees using an LC-MS/MS [18]. Clothianidin, imidacloprid, thiacloprid, thiamethoxam and nitenpyram were reported in honey, pollen and honeybees with serious contaminations of >150 ng/g for thiamethoxam in pollen, >70 ng/g for clothianidin in bees and 80 ng/g for a mixed 1:1 combination of the aforementioned neonicotinoids in honey. Based on the LOD of the nMIAs and the fortification experiments that are presented in Appendix A, we can conclude that the nMIAs can be successfully applied to those same samples. However, it should be emphasized that the fortification experiments that were below 10 ng/g were not performed, and that experiments were not performed on multiple neonicotinoid contaminations. Even though multi-contaminations will also be detected by the profiling assays, the classification of which neonicotinoids are present will become complex.

## 4. Conclusions

In this study, we successfully developed, evaluated, validated and applied five microsphere-based suspension immunoassays for the total neonicotinoid profiling of bee-related matrices. The five nMIAs were able to detect eight neonicotinoids at high sensitivities when they were compared to similar immunoassay approaches, including imidacloprid, clothianidin, thiacloprid and thiamethoxam which are prohibited for their outdoor use by the EU. To the best of our knowledge, this is the first broad-spectrum neonicotinoid profiling method that is based on immunoassays in the same instrumental format which are able to detect all of the commercially available neonicotinoids and its high sensitivities are promising when it is compared to those of other neonicotinoid detection formats. The chosen format is accepted worldwide as a reliable multiplexing format in diagnostic profiling applications. This guarantees the high stability and reliability of the microsphere image analyzer and its corresponding assay format. The advantages of these profiling assays are their rapidness, simplicity in the sample preparation, broad-spectrum detection, low cost and semi-high throughput. Additionally, the hapten-coupled microspheres are highly stable with average shelf lives of 1–2 years. Taken together, the nMIAs can be beneficial for both interest-based screening (e.g., apiaries) as well as (government) inspection services. The nMIAs were validated and applied to four matrices that can be easily sampled in apiaries and their environments. However, its application is not limited to apiaries and can, in general, be extended to the general screening of plants, feed and food. This was already successfully proven by the implementation of the Ace assay for screening a wide range of matrices in *Eucalyptus* sp. plantations [52]. Moreover, their implementation by plant retailers allows for stricter control in the “grower to customer” chain as both of them desire an effective phasing out of unwanted pesticides like neonicotinoids for more sustainable food and product chains. The use of a robust, transportable analyzer and the rapid and simple extraction qualities that it has show that there are good perspectives for acquiring measurements at the point of need with a very basic lab setup. Due to the specificity of the implemented antibodies, the profiling nMIAs will not always provide conclusive answers on which neonicotinoids are present in the case of multi-contaminations in a certain matrix which requires a follow-up confirmation by performing an LC-MS/MS. In conclusion, the implementation of the newly developed nMIAs contributes to a safer environment for bees and other essential pollinating insects.

## Figures and Tables

**Figure 1 biosensors-12-00792-f001:**
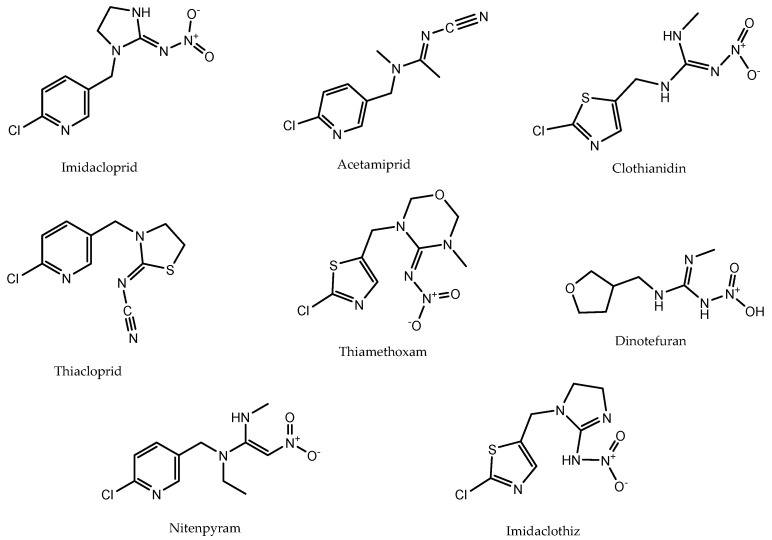
Chemical structures of the eight most relevant neonicotinoid pesticides.

**Figure 2 biosensors-12-00792-f002:**
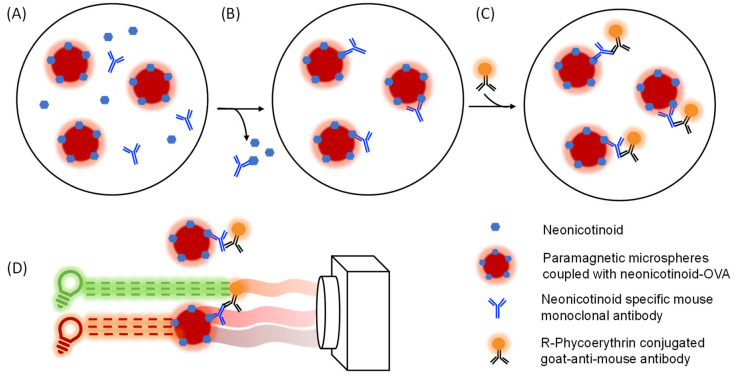
Assay and detection principle of the neonicotinoid microsphere immunoassays (nMIAs). OVA–neonicotinoid conjugates that are coupled microspheres are incubated with specific neonicotinoid antibodies and a sample containing free neonicotinoids. The OVA–neonicotinoid conjugates and free neonicotinoids compete in antibody binding (**A**). After incubation, the unbound antibodies and neonicotinoids are washed away (**B**). When it is incubated with an uncontaminated sample, the neonicotinoid antibody binds to the OVA–neonicotinoid conjugates on the microspheres, while fewer antibodies are bound to the microsphere when they are incubated with a neonicotinoid-contaminated sample. Next, an ample number of secondary anti-species antibodies that are coupled to R-phycoerythrin are incubated with the OVA-neonicotinoid-antibody complex and the excess is washed away (**C**). The microspheres contain two unique fluorochromes which emit red and far-red light upon excitation by a red LED. These emissions are captured using the CCD camera in the planar array reader for microsphere identification. Next, the intensity of the orange-emitted light from the reporter is measured after excitation by the green LED (**D**).

**Figure 3 biosensors-12-00792-f003:**
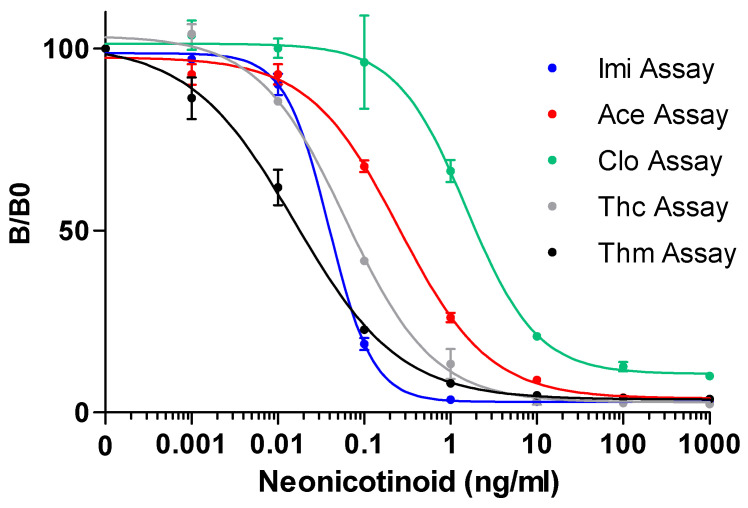
Dose-response curves in PBST-BSA buffer for the five neonicotinoid profiling assays (nMIAs) for detecting their corresponding neonicotinoid targets (*n* = 2).

**Figure 4 biosensors-12-00792-f004:**
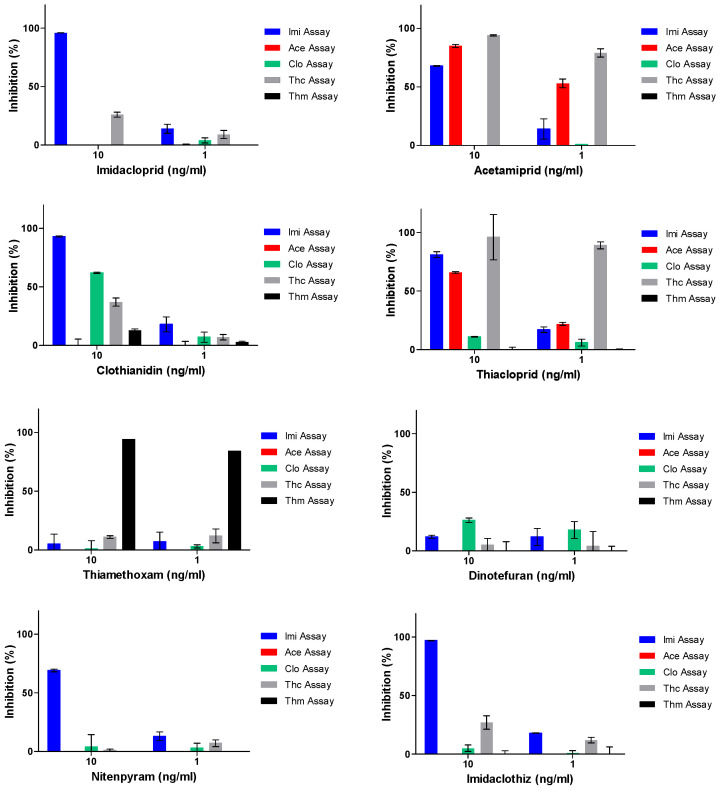
Inhibition profiles of the eight neonicotinoids in tap water at 10 and 1 ng/mL in the five neonicotinoid profiling assays (nMIAs) (*n* = 2).

**Table 1 biosensors-12-00792-t001:** Commercially available neonicotinoids.

Neonicotinoid	Abbreviation ^1^	Manufacturer(s) ^2^	Year Introduced ^3^
Imidacloprid	Imi	Bayer CropScience AG	1992
Thiacloprid	Thc	Bayer CropScience AG	2003
Thiamethoxam	Thm	Syngenta AG	1997
Nitenpyram	Nit	Sumitomo Chemical Takeda Agro Company	1995
Acetamiprid	Ace	Nippon Soda	1999
Clothianidin	Clo	Sumitomo Chemical Takeda Agro Company/Bayer CropScience AG	2003
Dinotefuran	Din	Mitsui Chemicals	2004
Imidaclothiz	Imc	Nantong Jiangshan Agrochemical & Chemical Co. Ltd.	2006

^1^ Abbreviations used in this article; ^2,3^ based on references [8,10,11,12,13].

**Table 2 biosensors-12-00792-t002:** Sensitivities (LODs) of the developed nMIAs and other neonicotinoid detection technologies from the literature.

Neonicotinoid	LOD	Method	Reference	
Imidacloprid	0.01 ng/mL	nMIA	Current study	
Acetamiprid	0.01 ng/mL			
Thiacloprid	0.02 ng/mL			
Clothianidin	0.02 ng/mL			
Thiamethoxam	0.003 ng/mL			
Dinotefuran	2.95 ng/mL			
Nitenpyram	0.09 ng/mL			
Imidaclothiz	0.07 ng/mL			
Imidacloprid	0.5 ng/mL	Quantum dots-based lateral flow immunoassay	Wang et al. 2017	[37]
Imidaclothiz	0.5 ng/mL			
Clothianidin	2.9 ng/mL			
Clothianidin	2.5 ng/mL	Recombinant antibody immunochromatographic assay	Chang et al. 2022	[43]
Thiamethoxam	79 ng/mL	Electrochemical monitoring platform using graphene carbon electrodes	Zhai et al. 2017	[28]
Thiamethoxam	0.5 ng/mL	Indirect competitive ELISA	Ye et al. 2018	[44]
Imidacloprid	1300 ng/mL	Molecular imprinted polymer-based sensor	El-Akaad et al. 2020	[30]
Imidacloprid	10 ng/mL	Rapid detection Quenchbody (scFv) assay	Zhao et al. 2018	[31]
Imidacloprid	0.03 ng/mL	Direct competitive ELISA	Kim et al. 2003	[45]
Acetamiprid	0.1 ng/mL	Quantum dots-based immunochromatographic assay	Liu et al. 2019	[36]
Acetamiprid	24 ng/mL	Fluorescent aptamer-based aggregation assay	Yang et al. 2018	[32]
Nitenpyram	7.3 ng/mL	Surface plasmon resonance immunosensor	Hirakawa et al. 2018	[46]
Dinotefuran	2090 ng/mL	Luminescence sensor by stable cadmium–organic framework	Jiao et al. 2021	[47]
Imidaclothiz	17.8 ng/mL	Indirect competitive ELISA	Fang et al. 2011	[48]
Thiacloprid	2.43 ng/mL	Fluorescence polarization immunoassay	Ding et al. 2019	[49]

**Table 3 biosensors-12-00792-t003:** Single nMIAs and their cross-reactive sensitivities (ng/mL) for all eight of the target neonicotinoids that are expressed in the limits of detection and IC_50_s.

Neonicotinoids	Imi Assay		Ace Assay		Clo Assay		Thc Assay		Thm Assay	
	LOD	IC_50_	LOD	IC_50_	LOD	IC_50_	LOD	IC_50_	LOD	IC_50_
Imidacloprid	**0.01**	**0.07**	-	101	-	-	0.15	41.7	-	-
Acetamiprid	0.07	0.81	**0.02**	**0.26**	-	-	0.01	0.19	-	-
Clothianidin	0.02	0.11	733	1107	**0.19**	**1.9**	2.32	29.5	0.45	32.0
Thiacloprid	0.05	0.47	0.66	2.1	-	-	**0.02**	**0.06**	-	-
Thiamethoxam	7.59	454.3	-	-	-	-	-	-	**0.003**	**0.01**
Dinotefuran	12.80	292	-	-	2.95	5.8	104.6	-	-	-
Nitenpyram	0.09	0.67	-	-	-	-	93.9	-	-	-
Imidaclothiz	0.04	0.11	-	-	-	-	2.7	23.88	-	-

“-“ means that the value is out of range, numbers in **bold** refer to the specific assay targets.

**Table 4 biosensors-12-00792-t004:** Overview of neonicotinoid profiling capabilities in pollen, honey and honeybees by the five nMIAs that were tested at 10 ng/g and 100 ng/g fortifications.

Neonicotinoid	Matrix	Imi Assay	Ace Assay	Clo Assay	Thc Assay	Thm Assay
Imidacloprid	Pollen	10 ng/g			100 ng/g	
	Honey	10 ng/g			100 ng/g	
	Bee	10 ng/g			100 ng/g	
Acetamiprid	Pollen	100 ng/g	10 ng/g		10 ng/g	
	Honey	100 ng/g	10 ng/g		10 ng/g	
	Bee	100 ng/g	10 ng/g		10 ng/g	
Clothianidin	Pollen	10 ng/g		100 ng/g		
	Honey	10 ng/g		100 ng/g		
	Bee	10 ng/g		100 ng/g		
Thiacloprid	Pollen	100 ng/g	10 ng/g		10 ng/g	
	Honey	100 ng/g	10 ng/g		10 ng/g	
	Bee	100 ng/g	10 ng/g		10 ng/g	
Thiamethoxam	Pollen					10 ng/g
	Honey					10 ng/g
	Bee					10 ng/g
Dinotefuran	Pollen			100 ng/g		
	Honey			100 ng/g		
	Bee			100 ng/g		
Nitenpyram	Pollen	10 ng/g				
	Honey	10 ng/g				
	Bee	10 ng/g				
Imidaclothiz	Pollen	10 ng/g				
	Honey	10 ng/g				
	Bee	10 ng/g				

## Data Availability

Data is contained within the article or Appendix A, additional data is available from the corresponding author upon request.

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
