# Peer review of "Neonicotinoid Microsphere Immunosensing for Profiling Applications in Honeybees and Bee-Related Matrices"

_biosensors, 2022, doi:10.3390/bios12100792_

Round 1

Reviewer 1 Report

Review of manuscript number biosensors-1865416

The received manuscript entitled Neonicotinoid microsphere immunosensing for profiling applications in honeybees and bee-related matrices provides an interesting microsphere-based suspension immunoassays for neonicotinoid profiling and their application to bees and essential bee-related matrices. These five single neonicotinoid microsphere immunoassays (nMIAs) have been developed on xMAP technology, which enabled the detection of imidacloprid, acetamiprid, clothianidin, thiamethoxam, thiacloprid, dinotefuran, nitenpyram and imidaclothiz belong to neonicotinoids. Proposed assays detect some neonicotinoids (detailly five) that are currently banned by the EU and another three neonicotinoids which are not released for the European market. The developed assays were applied to matrices relevant to the honeybee environment (water, pollen, honey) and honeybees directly. Proposed methods of extraction were implemented and resulted in the successful detection of all eight neonicotinoids in all matrices at low ng/mL levels. The combination of these assays allows the total profiling of all eight common neonicotinoids in bee-related matrices.  

Remarks:

In Abstract section more substantial information including qualitative results are needed.

The introduction contains only minimum information about the issue under discussion like analytical methods. It should be rewritten to highlight the purpose of investigations.

The critical evaluation of more important quantitative merits (such as LOD, LOQ, sensitivity, accuracy, etc) is required.

The novelty and significance of proposed investigations require critical evaluation and should be discussed.

Any scientific work should contain important elements of novelty and a new approach to the discussed problems. What are the significant differences from other scientific works in this field? Moreover, please identify advantages and disadvantages of the proposed methodology, discuss the possibilities of the obtained results use and what are the outlooks for the future?

 Summary:

The subject of the manuscript falls within the scope of Biosensors. The minor revision of the manuscript is needed and after revision it is recommended for the publication.

Author Response

Reviewer 1:

We would like thank reviewer 1 for the valuable comments on our manuscript. Our response can be found below.

Note: for cross-reference, line numbers relate to the revised document in “all mark up” mode.

  1. In Abstract section more substantial information including qualitative results are needed.

We do agree with the reviewer and have added the LODs for all the developed assays to the abstract.

  1. The introduction contains only minimum information about the issue under discussion like analytical methods. It should be rewritten to highlight the purpose of investigations.

We do agree with the reviewer and have extended the introduction on detection methods for neonicotinoids, including new references (L97-L113).

  1. The critical evaluation of more important quantitative merits (such as LOD, LOQ, sensitivity, accuracy, etc) is required.

We do agree with the reviewer and have therefore added a new table: Sensitivities (LODs) of the developed nMIAs and other neonicotinoid detection technologies from literature (L380) for critical evaluation. Additionally, we have added 2 new sections where we compare the application of our method to matrix samples with values found in surveys in naturally contaminated materials (L459-L466) and (L538-L555).

  1. The novelty and significance of proposed investigations require critical evaluation and should be discussed.
  2. Any scientific work should contain important elements of novelty and a new approach to the discussed problems. What are the significant differences from other scientific works in this field? Moreover, please identify advantages and disadvantages of the proposed methodology, discuss the possibilities of the obtained results use and what are the outlooks for the future?

We agree with the reviewer on points 4 and 5 and have therefore seriously expanded the conclusion section to better highlight the significance and novelty of the proposed investigations (L564-L602). Additionally, we have added a new section to the introduction (L146-L151).

Reviewer 2 Report

In this manuscript, a series of neonicotinoid microsphere immunoassays were developed for the detection of eight neonicotinoids insecticides, and were applied to fortified matrices including surface water, pollen, honey and honeybees. The manuscript is well organized and the result is interesting. However, it need minor revision before the publication in Biosensors.

1. There are some other techniques that were commonly used for the detection of pesticide residue, such as fluorescent assays and electrochemical assays. Thus, their comparison with immunoassays should be added in the introduction part.

2. The sensitivity is vital for a new analytical method. In the reviewers mind, it should be better to provide a table to compare the sensitivity of the above mentioned assay for  neonicotinoids detection.

3. In addition to the sensitivity, some other factors may also affect the feasibility of the developed method for practical use, such as the time and money cost. Please provide essential discussion about this issue.

4. Some grammatical errors exist in the manuscript.

Author Response

We would like thank reviewer 2 for the valuable comments on our manuscript. Our response can be found below.

Reviewer 2

Note: for cross-reference, line numbers relate to the revised document in “all mark up” mode.

  1. There are some other techniques that were commonly used for the detection of pesticide residue, such as fluorescent assays and electrochemical assays. Thus, their comparison with immunoassays should be added in the introduction part.

We agree with the reviewer and have therefore extended this section in the introduction (L97-L113), including relevant references.

  1. The sensitivity is vital for a new analytical method. In the reviewer’s mind, it should be better to provide a table to compare the sensitivity of the above mentioned assay for neonicotinoids detection.

We do agree with the reviewer and have therefore added a new table: Sensitivities (LODs) of the developed nMIAs and other neonicotinoid detection technologies from literature (L380) for critical evaluation. Additionally, we have added 2 new sections where we compare the application of our method to matrix samples with values found in surveys in naturally contaminated materials (L459-L466) and (L538-L555).

  1. In addition to the sensitivity, some other factors may also affect the feasibility of the developed method for practical use, such as the time and money cost. Please provide essential discussion about this issue.

We agree with the reviewer and have therefore seriously expanded the conclusion section to better highlight the significance and novelty of the proposed investigations (L564-L602). Additionally, we have added the following information (L405-L406):

For the nMIAs,  preparation and measurement of 96 samples is achieved within 2 hours.

  1. Some grammatical errors exist in the manuscript.

The manuscript has been critically controlled and grammatical changes were introduced.

Reviewer 3 Report

The idea presented in the manuscript is interesting, with a relevant perspective on EU legislation. However, some changes/modifications are required to improve the quality of the work.

Major revisions

1-The abstract does not clearly indicate the technique/methodology used to carry out the detection and/or how the analytical signal is obtained.

2- L183 Section 2.4 Antibody dilution factor optimization is hard to understand. Authors should rewrite succinctly.

In fact, the "Materials and Methods" section is extensive and complex to understand. I suggest the authors to review this section and make it clearer.

3- Why is OVA used? What is its main effect/benefits? Couldn't another protein be used? This information must be added to the article. Also, why is BSA added to the negative controls (L204)? Is the OVA not enough? Can OVA be replaced for BSA?

4- Different material (tap water, pollen, honey, or bees) was tested. The material is provided from different countries/cities. Are the results correlated? Authors should mention in the manuscript why those materials were chosen and what the expected/ the correlation between the obtained results. It should be clarified, mainly in the introduction part.

5- Legend of Figure S3 - Please indicate what the solid and dashed lines mean and how those values were obtained.

6- A comparison of the results with other studies for the detection of neonicotinoids in tap water, honey etc is not described. Authors should briefly describe the advantage of this work over others.

Minor revisions

*Table 1 and Figure 1 should be added in the results discussion part or in the supporting information. I don't think it's relevant in the introduction.

*L58. Authors refer that certain neonicotinoids are not harmful to bee colonies. Please refer specifically to the neonicotinoids. Also, reduce/make information clearer between L58-L61.

*L354 - remove the comma before the dot.

Author Response

We would like to thank reviewer 3 for the valuable comments on our manuscript. Please find our responses below.

Note: for cross-reference, line numbers relate to the revised document in “all mark up” mode.

1-The abstract does not clearly indicate the technique/methodology used to carry out the detection and/or how the analytical signal is obtained.

We agree with the reviewer and have added this information to the abstract (L15-L18)

“The presented study describes the development of novel competitive, fluorescent microsphere-based suspension immunoassays for neonicotinoid profiling and their application to bees and essential bee-related matrices, using the multi analyte profiling (xMAP) technology”

2- L183 – Section “2.4 Antibody dilution factor optimization” is hard to understand. Authors should rewrite succinctly.

This section was rewritten and is now hopefully easier to understand for the reviewer (L223-L248):

In fact, the "Materials and Methods" section is extensive and complex to understand. I suggest the authors to review this section and make it clearer.

Modifications have been made throughout the materials/methods section that we believe will contribute to a better understanding of this section.

3- Why is OVA used? What is its main effect/benefits? Couldn't another protein be used? This information must be added to the article. Also, why is BSA added to the negative controls (L204)? Is the OVA not enough? Can OVA be replaced for BSA?

BSA and OVA are generic carrier proteins that are commonly used for coupling small molecules (haptens), to make them suitable antigens for immunoassays. In our case, the hapten-OVA conjugates are covalently coupled to the microspheres. Companies normally provide either BSA and/or OVA hapten conjugates. Information was already provided at the start of the discussion, but we have now added additional information (L322-L324)

“Neonicotinoids are small molecules (haptens) that require coupling to larger carrier proteins in order to be used as coating antigens in immunoassays. Both OVA and BSA are commonly used for the generation of these hapten-protein conjugates”

Additionally, the BSA is not added to the control (or samples), but is present in the buffer and functions as a blocking agent, to avoid non-specific interactions of matrix components. It is present in all the samples. Blocking could also be achieved by using OVA or milk powder in the buffer. We do understand the confusion and therefore have changed (L256)

“Negative control standards only contained the PBST-BSA assay buffer”

To

“PBST-BSA buffer without fortifications was used as a negative control in each assay”

4- Different material (tap water, pollen, honey, or bees) was tested. The material is provided from different countries/cities. Are the results correlated? Authors should mention in the manuscript why those materials were chosen and what the expected/ the correlation between the obtained results. It should be clarified, mainly in the introduction part.

We do agree with the reviewer that the choice of the materials was not clearly explained in the manuscript and therefore we have added the following section to the introduction (L146-L151).

The conducted research is part of the European H2020 B-GOOD project ((https://b-good-project.eu/) that develops new diagnostic tools that help to map the health status of honeybee colonies and their environment, to give beekeeping guidance by computational-assisted decision making. Therefore we chose to focus on essential resources for bees (water, pollen) and matrices relevant as health-based indicators to the bee environment (bees, honey) in the presented research.

Water samples are collected from different waterbodies in order to include matrices with different composition of ions, minerals and microorganisms for the application and validation of the nMIAs. We have also addressed the comment considering the water samples that originated from different cities/countries in the discussion (L513-L516):

“Even though the water samples originated from very diverse water bodies, the validation results for the nMIAs (figure S3) show that compositional differences (e.g. minerals) did not influence the assays at the critical concentrations assessed”.

5- Legend of Figure S3 - Please indicate what the solid and dashed lines mean and how those values were obtained.

We do agree with the reviewer that this information is lacking and have therefore adjusted the caption of Figure S3 in the supporting information accordingly.

6- A comparison of the results with other studies for the detection of neonicotinoids in tap water, honey etc is not described. Authors should briefly describe the advantage of this work over others.

We do agree with the reviewer and therefore have added 2 new sections where we compare the application of our method to matrix samples with values found in surveys in naturally contaminated materials (L459-L466) and (L538-L555). Besides that we have added a new table: Sensitivities (LODs) of the developed nMIAs and other neonicotinoid detection technologies from literature (L380) for additional critical evaluation.

Minor revisions

*Table 1 and Figure 1 should be added in the results discussion part or in the supporting information. I don't think it's relevant in the introduction.

Only in this case, we do not agree with the reviewer. The inclusion in the introduction helps readers to realize that the manuscript deals with the detection of low molecular weight compounds. Additionally, since neonicotinoids and their respective detrimental effects on the bees are an important factor for the proposed research (as now clearly stated in the introduction), we would like to keep this part in the introduction

*L58. Authors refer that “certain neonicotinoids are not harmful to bee colonies”. Please refer specifically to the neonicotinoids. Also, reduce/make information clearer between L58-L61.

We agree with the reviewer that this information needs a better specification, therefore this section was changed to:

“Although several studies showed that neonicotinoids including imidacloprid, clothianidin and thiamethoxam are not lethal to bees at field-realistic concentrations, the collective exposure of these neonicotinoids can still affect bees’ health, stress and homing ability” (L61-L66)

*L354 - remove the comma before the dot

Done
